# Fast Determinantal Point Process Sampling with Application to Clustering

**Byungkon Kang** *
Samsung Advanced Institute of Technology
Yongin, Korea
bk05.kang@samsung.com

## Abstract

Determinantal Point Process (DPP) has gained much popularity for modeling sets of diverse items. The gist of DPP is that the probability of choosing a particular set of items is proportional to the determinant of a positive definite matrix that defines the similarity of those items. However, computing the determinant requires time cubic in the number of items, and is hence impractical for large sets. In this paper, we address this problem by constructing a rapidly mixing Markov chain, from which we can acquire a sample from the given DPP in sub-cubic time. In addition, we show that this framework can be extended to sampling from cardinality-constrained DPPs. As an application, we show how our sampling algorithm can be used to provide a fast heuristic for determining the number of clusters, resulting in better clustering.

<span style="color:red">There are some crucial errors in the proofs of the theorem which invalidate the theoretical claims of this paper. Please consult the appendix for more details.</span>

## 1 Introduction

Determinantal Point Process (DPP) [1] is a well-known framework for representing a probability distribution that models diversity. Originally proposed to model repulsion among physical particles, it has found its way into many applications in AI, such as image search [2] and text summarization [3].

In a nutshell, given an itemset $S = [n] = \{1, 2, \cdots, n\}$ and a symmetric positive definite (SPD) matrix $L \in \mathbb{R}^{n \times n}$ that describes pairwise similarities, a (discrete) DPP is a probability distribution over $2^S$ proportional to the determinant of the corresponding submatrix of $L$. It is known that this distribution assigns more probability mass on set of points that have larger diversity, which is quantified by the entries of $L$.

Although the size of the support is exponential, DPP offers tractable inference and sampling algorithms. However, sampling from a DPP requires $O(n^3)$ time, as an eigen-decomposition of $L$ is necessary [4]. This presents a huge computational problem when there are a large number of items; *e.g.*, $n > 10^4$. A motivating problem we consider is that of kernelized clustering [5]. In this problem, we are given a large number of points plus a kernel function that serves as a dot product between the points in a feature space. The objective is to partition the points into some number clusters, each represented by a point called *centroid*, in a way that a certain cost function is minimized. Our approach is to sample the centroids via DPP. This heuristic is based on the fact that each cluster should be different from one another as much as possible, which is precisely what DPPs prefer. Naively using the cubic-complexity sampling algorithm is inefficient, since it can take up to a whole day to eigen-decompose a $10000 \times 10000$ matrix.

In this paper, we present a rapidly mixing Markov chain whose stationary distribution is the DPP

of interest. Our Markov chain does not require the eigen-decomposition of $L$, and is hence time-efficient. Moreover, our algorithm works seamlessly even when new items are added to $S$ (and $L$), while the previous sampling algorithm requires expensive decompositions whenever $S$ is updated.

## 1.1 Settings

More specifically, a DPP over the set $S = [n]$, given a positive-definite similarity matrix $L \succ \mathbf{0}$, is a probability distribution $P_L$ over any $Y \subseteq S$ in the following form:

$$P_L(\mathbf{Y} = Y) = \frac{\det(L_Y)}{\sum_{Y' \subseteq S} \det(L_{Y'})} = \frac{\det(L_Y)}{\det(L + I)},$$

where $I$ is the identity matrix of corresponding dimension, $\mathbf{Y}$ is a random subset of $S$, and $L_Y \succ \mathbf{0}$ is the principal minor of $L$ whose rows and columns are restricted to the elements of $Y$. *i.e.*, $L_Y = [L(i,j)]_{i,j \in Y}$, where $L(i,j)$ is the $(i,j)$ entry of $L$. Many literatures introduce DPP in terms of a *marginal kernel* that describes marginal probabilities of inclusion. However, since directly modeling probabilities over each subset of $S^1$ offers a more flexible framework, we will focus on the latter representation.

There is a variant of DPPs that places a constraint on the size of the random subsets. Given an integer $k$, a $k$-DPP is a DPP over size-$k$ sets [2]:

$$P_L^k(\mathbf{Y} = Y) = \begin{cases} \frac{\det(L_Y)}{\sum_{|Y'| = k} \det(L_{Y'})}, & \text{if } |Y| = k \\ 0, & \text{otherwise.} \end{cases}$$

During the discussions, we will use a *characteristic vector* representation of each $Y \subseteq S$; *i.e.*, $v_Y \in \{0,1\}^{|S|}, \forall Y \subseteq S$, such that $v_Y(i) = 1$ if $i \in Y$, and 0 otherwise. With abuse of notation, we will often use set operations on characteristic vectors to indicate the same operation on the corresponding sets. *e.g.*, $v_Y \setminus \{u\}$ is equivalent to setting $v_Y(u) = 0$ and correspondingly, $Y \setminus \{u\}$.

## 2 Algorithm

The overall idea of our algorithm is to design a rapidly-mixing Markov chain whose stationary distribution is $P_L$. The state space of our chain consists of the characteristic vectors of each subset of $S$. This Markov chain is generated by a standard Metropolis-Hastings algorithm, where the transition probability from state $v_Y$ to $v_Z$ is given as the ratio of $P_L(Z)$ to $P_L(Y)$. In particular, we will only consider transitions between adjacent states - states that have Hamming distance 1. Hence, the transition probability of removing an element $u$ from $Y$ is of the following form:

$$\Pr(Y \cup \{u\} \to Y) = \min \left\{ 1, \frac{\det(L_Y)}{\det(L_{Y \cup \{u\}})} \right\}.$$

The addition probability is defined similarly. The overall chain is an insertion/deletion chain, where a uniformly proposed element is either added to, or subtracted from the current state. This procedure is outlined in Algorithm 1. Note that this algorithm has a potentially high computational complexity, as the determinant of $L_Y$ for a given $Y \subseteq S$ must be computed on every iteration. If the size of $Y$ is large, then a single iteration will become very costly. Before discussing how to address this issue in Section 2.1, we analyze the properties of Algorithm 1 to show that it efficiently samples from $P_L$. First, we state that the chain induced by Algorithm 1 does indeed represent our desired distribution[2].

**Proposition 1.** *The Markov chain in Algorithm 1 has a stationary distribution $P_L$.*

The computational complexity of sampling from $P_L$ using Algorithm 1 depends on the *mixing time* of the Markov chain; *i.e.*, the number of steps required in the Markov chain to ensure that the current distribution is "close enough" to the stationary distribution. More specifically, we are interested in the $\epsilon$-mixing time $\tau(\epsilon)$, which guarantees a distribution that is $\epsilon$-close to $P_L$ in terms of total variation. In other words, we must spend at least this many time steps in order to acquire a sample from a distribution close to $P_L$. Our next result states that the chain in Algorithm 1 mixes rapidly:

**Algorithm 1** Markov chain for sampling from $P_L$
---
**Require:** Itemset $S = [n]$, similarity matrix $L \succ \mathbf{0}$
  Randomly initialize state $Y \subseteq S$
  **while** Not mixed **do**
    Sample $u \in S$ uniformly at random
    Set

$$p_u^+(Y) \leftarrow \min \left\{ 1, \frac{\det(L_{Y \cup \{u\}})}{\det(L_Y)} \right\}$$

$$p_u^-(Y) \leftarrow \min \left\{ 1, \frac{\det(L_{Y \setminus \{u\}})}{\det(L_Y)} \right\}$$

    **if** $u \notin Y$ **then**
      $Y \leftarrow Y \cup \{u\}$ with prob. $p_u^+(Y)$
    **else**
      $Y \leftarrow Y \setminus \{u\}$ with prob. $p_u^-(Y)$
    **end if**
  **end while**
  **return** $Y$
---

**Theorem 1.** *The Markov chain in Algorithm 1 has a mixing time $\tau(\epsilon) = O\left(n \log\left(n/\epsilon\right)\right)$.*

One advantage of having a rapidly-mixing Markov chain as means of sampling from a DPP is that it is robust to addition/deletion of elements. That is, when a new element is introduced to or removed from $S$, we may simply continue the current chain until it is mixed again to obtain a sample from the new distribution. Previous sampling algorithm, on the other hand, requires to expensively eigen-decompose the updated $L$.

The mixing time of the chain in Algorithm 1 seems to offer a promising direction for sampling from $P_L$. However, note that this is subject to the presence of an efficient procedure for computing $\det(L_Y)$. Unfortunately, computing the determinant already costs $O(|Y|^3)$ operations, rendering Algorithm 1 impractical for large $Y$'s. In the following sections, we present a linear-algebraic manipulation of the determinant ratio so that explicit computation of the determinants is unnecessary.

## 2.1 Determinant Ratio Computation

It turns out that we do not need to explicitly compute the determinants, but rather the ratio of determinants. Suppose we wish to compute $\det(L_{Y \cup \{u\}}) / \det(L_Y)$. Since the determinant is permutation-invariant with respect to the index set, we can represent $L_{Y \cup \{u\}}$ as the following block matrix form, due to its symmetry:

$$L_{Y \cup \{u\}} = \begin{pmatrix} L_Y & b_u \\ b_u^\top & c_u \end{pmatrix},$$

where $b_u = (L(i, u))_{i \in Y} \in \mathbb{R}^{|Y|}$ and $c_u = L(u, u)$. With this, the determinant of $L_{Y \cup \{u\}}$ is expressed as

$$\det(L_{Y \cup \{u\}}) = \det(L_Y) \left( c_u - b_u^\top L_Y^{-1} b_u \right). \tag{1}$$

This allows us to re-formulate the insertion transition probability as a determinant-free ratio.

$$p_u^+(Y) = \min \left\{ 1, \frac{\det(L_{Y \cup \{u\}})}{\det(L_Y)} \right\} = \min \left\{ 1, c_u - b_u^\top L_Y^{-1} b_u \right\}. \tag{2}$$

The deletion transition probability $p_u^-(Y \cup \{u\})$ is computed likewise:

$$p_u^-(Y \cup \{u\}) = \min \left\{ 1, \frac{\det(L_Y)}{\det(L_{Y \cup \{u\}})} \right\} = \min \left\{ 1, (c_u - b_u^\top L_Y^{-1} b_u)^{-1} \right\}.$$

However, this transformation alone does not seem to result in enhanced computation time, as computing the inverse of a matrix is just as time-consuming as computing the determinant.

To save time on computing $L_{Y'}^{-1}$, we incrementally update the inverse through blockwise matrix inversion. Suppose that the matrix $L_Y^{-1}$ has already been computed at the current iteration of the chain. First, consider the case when an element $u$ is added ('if' clause). The new inverse $L_{Y \cup \{u\}}^{-1}$ must be updated from the current $L_Y^{-1}$. This is achieved by the following block-inversion formula [6]:

$$L_{Y \cup \{u\}}^{-1} = \begin{pmatrix} L_Y & b_u \\ b_u^\top & c_u \end{pmatrix}^{-1} = \begin{pmatrix} L_Y^{-1} + L_Y^{-1} b_u b_u^\top L_Y^{-1}/d_u & -L_Y^{-1} b_u/d_u \\ -b_u^\top L_Y^{-1}/d_u & d_u \end{pmatrix}, \tag{3}$$

where $d_u = c_u - b_u^\top L_Y^{-1} b_u$ is the Schur complement of $L_Y$. Since $L_Y^{-1}$ is already given, computing each block of the new inverse matrix costs $O(|Y|^2)$, which is an order faster than the $O(|Y|^3)$ complexity required by the determinant. Moreover, only half of the entries may be computed, due to symmetry.

Next, consider the case when an element $u$ is removed ('else' clause) from the current set $Y$. The matrix to be updated is $L_{Y \setminus \{u\}}^{-1}$, and is given by the rank-1 update formula. We first represent the current inverse $L_Y^{-1}$ as

$$L_Y^{-1} = \begin{pmatrix} L_{Y \setminus \{u\}} & b_u \\ b_u^\top & c_u \end{pmatrix}^{-1} \triangleq \begin{pmatrix} D & e \\ e^\top & f \end{pmatrix},$$

where $D \in \mathbb{R}^{(|Y|-1) \times (|Y|-1)}$, $e \in \mathbb{R}^{|Y|-1}$, and $f \in \mathbb{R}$. Then, the inverse of the submatrix $L_{Y \setminus \{u\}}$ is given by

$$L_{Y \setminus \{u\}}^{-1} = D - \frac{ee^\top}{f}. \tag{4}$$

Again, updating $L_{Y \setminus \{u\}}^{-1}$ only requires matrix arithmetic, and hence costs $O(|Y|^2)$.

However, the initial $L_Y^{-1}$, from which all subsequent inverses are updated, must be computed in full at the beginning of the chain. This complexity can be reduced by restricting the size of the initial $Y$. That is, we first randomly initialize $Y$ with a small size, say $o(n^{1/3})$, and compute the inverse $L_Y^{-1}$. As we proceed with the chain, update $L_Y^{-1}$ using Equations 3 and 4 when an insertion or a deletion proposal is accepted, respectively. Therefore, the average complexity of acquiring a sample from a distribution that is $\epsilon$-close to $P_L$ is $O(T^2 n \log(n/\epsilon))$, where $T$ is the average size of $Y$ encountered during the progress of chain. In Section 3, we introduce a scheme to maintain a small-sized $Y$.

## 2.2 Extension to $k$-DPPs

The idea of constructing a Markov chain to obtain a sample can be extended to $k$-DPPs. The only known algorithm so far for sampling from a $k$-DPP also requires the eigen-decomposition of $L$. Extending the previous idea, we provide a Markov chain sampling algorithm similar to Algorithm 1 that samples from $P_L^k$.

The main idea behind the $k$-DPP chain is to propose a new configuration by choosing two elements: one to remove from the current set, and another to add. We accept this move according to the probability defined by the ratio of the proposed determinant to the current determinant. This is equivalent to selecting a row and column of $L_X$, and replacing it with the ones corresponding to the element to be added. *i.e.*, for $X = Y \cup \{u\}$

$$L_{X=Y \cup \{u\}} = \begin{pmatrix} L_Y & b_u \\ b_u^\top & c_u \end{pmatrix} \Rightarrow L_{X'=Y \cup \{v\}} = \begin{pmatrix} L_Y & b_v \\ b_v^\top & c_v \end{pmatrix},$$

where $u$ and $v$ are the elements being removed and added, respectively. Following Equation 2, we set the transition probability as the ratio of the determinants of the two matrices.

$$\frac{\det(L_{X'})}{\det(L_X)} = \frac{c_v - b_v^\top L_Y^{-1} b_v}{c_u - b_u^\top L_Y^{-1} b_u}.$$

The final procedure is given in Algorithm 2.

Similarly to Algorithm 1, we present the analysis on the stationary distribution and the mixing time of Algorithm 2.

**Proposition 2.** *The Markov chain in Algorithm 2 has a stationary distribution $P_L^k$.*

---
**Algorithm 2** Markov chain for sampling from $P_L^k$

---
**Require:** Itemset $S = [n]$, similarity matrix $L \succ \mathbf{0}$
  Randomly initialize state $X \subseteq S$, s.t. $|X| = k$
  **while** Not mixed **do**
    Sample $u \in X$, and $v \in S \setminus X$ u.a.r.
    Letting $Y = X \setminus \{u\}$, set

$$p \leftarrow \min \left\{ 1, \frac{c_v - b_v^\top L_Y^{-1} b_v}{c_u - b_u^\top L_Y^{-1} b_u} \right\}. \tag{5}$$

    $X \leftarrow Y \cup \{v\}$ with prob. $p$
  **end while**
  **return** $X$

---

**Theorem 2.** *The Markov chain in Algorithm 2 has a mixing time $\tau(\epsilon) = O(k \log(k/\epsilon))$.*

The main computational bottleneck of Algorithm 2 is the inversion of $L_Y$. Since $L_Y$ is a $(k-1) \times (k-1)$ matrix, the per-iteration cost is $O(k^3)$. However, this complexity can be reduced by applying Equation 3 on every iteration to update the inverse. This leads to the final sampling complexity of $O(k^3 \log(k/\epsilon))$, which dominates the $O(k^3)$ cost of computing the intitial inverse, for acquiring a single sample from the chain. In many cases, $k$ will be a constant much smaller than $n$, so our algorithm is efficient in general.

## 3 Application to Clustering

Finally, we show how our algorithms lead to an efficient heuristic for a $k$-means clustering problem when the number of clusters is not known. First, we briefly overview the $k$-means problem.

Given a set of points $P = \{x_i \in \mathbb{R}^d\}_{i=1}^n$, the goal of clustering is to construct a partition $C = \{C_1, \cdots, C_k | C_i \subseteq P\}$ of $P$ such that the *distortion*

$$D_C = \sum_{i=1}^k \sum_{x \in C_i} \|x - m_i\|_2^2 \tag{6}$$

is minimized, where $m_i$ is the *centroid* of cluster $C_i$. It is known that the optimal centroid is the mean of the points of $C_i$. *i.e.*, $m_i = (\sum_{x \in C_i} x)/|C_i|$. Iteratively minimizing this expression converges to a local optimum, and is hence the preferred approach in many works. However, determining the number of clusters $k$ is the factor that makes this problem NP-hard [7]. Note that the problem of unknown $k$ prevails in other types of clustering algorithm, such as kernel $k$-means [5] and spectral clustering [8]: Kernel $k$-means is exactly the same as regular $k$-means except that the inner-products are substituted with a positive semi-definite kernel function, and spectral clustering uses regular $k$-means clustering as a subroutine. Some common techniques to determine $k$ include performing a density-based analysis of the data [9], or selecting $k$ that minimizes the Bayesian information criterion (BIC) [10].

In this work, we propose to sample the initial centroids of the clustering via our DPP sampling algorithms. The similarity matrix $L$ will be the Gram matrix determined by $L(i,j) = \kappa(x_i, x_j)$, where $\kappa(\cdot)$ is simply the inner-product for regular $k$-means, and a specified kernel function for kernel $k$-means. Since DPPs naturally capture the notion of diversity, the sampled points will tend to be more diverse, and thus serve better as initial representatives for each cluster. Once we have a sample, we perform a Voronoi partition around the elements of the sample to obtain a clustering[3]. Note that it is not necessary to determine $k$ beforehand as it can be obtained from the DPP samples. This approach is closely related to the MAP inference problem for DPPs [11], which is known to be NP-Hard as well. We use the proposed algorithms to sample the representative points that have high probability under $P_L$, and cluster the rest of the points around the sample. Subsequently, standard (kernel) $k$-means algorithms can be applied to improve this initial clustering.

Since DPPs model both size and diversity, it seems that we could simply collect samples from Algorithm 1 directly, and use those samples as representatives. However, as pointed out by [2], modeling both properties simultaneously can negatively bias the quality of diversity. To reduce this possible negative influence, we adopt a two-step sampling strategy: First, we gather $C$ samples from Algorithm 1 and construct a histogram $H$ over the sizes of the samples. Next, we sample from $k$-DPPs, by Algorithm 2, on a $k$ acquired from $H$. This last sample is the representatives we use to cluster.

Another problem we may encounter in this scheme is the sensitivity to outliers. The presence of an outlier in $P$ can cause the DPP in the first phase to favor the inclusion of that outlier, resulting in a possibly bad clustering. To make our approach more robust to outliers, we introduce the following *cardinality-penalized DPP*:

$$P_{L;\lambda}(\mathbf{Y} = Y) \propto \exp(\text{tr}(\log(L_Y)) - \lambda|Y|) = \frac{\det(L_Y)}{\exp(\lambda|Y|)},$$

where $\lambda \geq 0$ is a hyper-parameter that controls the weight to be put on $|Y|$. This regularization scheme smoothes the original $P_L$ by exponentially discounting the size of $Y$'s. This does not increase the order of the mixing time of the induced chain, since only a constant factor of $\exp(\pm\lambda)$ is multiplied to the transition probabilities. Algorithm 3 describes the overall procedure of our DPP-based clustering.

---

**Algorithm 3** DPP-based Clustering

**Require:** $L \succ \mathbf{0}, \lambda \geq 0, R > 0, C > 0$
    Gather $\{S_1, \cdots, S_C | S_i \sim P_{L;\lambda}\}$ (Algorithm 1)
    Construct histogram $H = \{|S_i|\}_{i=1}^C$ on the sizes of $S_i$'s
    **for** $j = 1, \cdots, R$ **do**
        Sample $M_j \sim P_L^{k_j}$ (Algorithm 2), where $k_j \sim H$
        Voronoi partition around $M_j$
    **end for**
    **return** clustering with lowest distortion (Equation 6)

---

Choosing the right value of $\lambda$ usually requires a priori knowledge of the data set. Since this information is not always available, one may use a small subset of $P$ to heuristically choose $\lambda$. For example, examine the BIC of the initial clustering with respect to the centroids sampled from $O(\sqrt{n})$ randomly chosen elements $P' \subset P$, with $\lambda = 0$. Then, increase $\lambda$ by 1 until we encounter the point where the BIC hits the local maximum to choose the final value. An additional binary search step may be used between $\lambda$ and $\lambda + 1$ to further fine-tune its value. Because we only use $O(\sqrt{n})$ points to sample from the DPP, each search step has at most linear complexity, allowing for ample time to choose better $\lambda$'s. This procedure may not appear to have an apparent advantage over a standard BIC-based model selection to choose the number of clusters $k$. However, tuning $\lambda$ not only allows one to determine $k$, but also gives better initial partitions in terms of distortion.

## 4 Experiments

In this section, we empirically demonstrate how our proposed method, denoted DPP-MC, of choosing an initial clustering compares to other methods, in terms of distortion and running time. The methods we compare against include:

- DPP-Full: Sample using full DPP sampling procedure as given in [4].
- DPP-MAP: Sample the initial centroids according to the MAP configuration, using the algorithm of [11].
- KKM: Plain kernel $k$-means clustering given by [5], run on the "true" number of clusters.

DPP-Full and DPP-MAP were used only in the first phase of Algorithm 3. To summarize the testing procedure, DPP-MC, DPP-Full, DPP-MAP were used to choose the initial centroids. After this initialization, KKM was carried out to improve the initial partitioning. Hence, the only difference between the algorithms tested and KKM is the initialization.

The real-world data sets we use are the letter recognition data set [12] (LET), and a subset of the power consumption data set [13] (PWC), The LET set is represented as 10,000 points in $\mathbb{R}^{16}$, and the PWC set 10,000 points in $\mathbb{R}^7$. While the LET set has 26 ground-truth clusters, the PWC set is only labeled with timestamps. Hence, we manually divided all points into four clusters, based on the month of timestamps. Since this partitioning is not the ground truth given by the data collector, we expected the KKM algorithm to perform badly on this set.

In addition, we also tested our algorithm on an artificially-generated set consisting of 15,000 points in $\mathbb{R}^{10}$ from five mixtures of Gaussians (MG). However, this task is made challenging by roughly merging the five Gaussians so that it is more likely to discover fewer clusters. The purpose of this set is to examine how well our algorithm finds the appropriate number of clusters. For the MG set, we present the result of DBSCAN [9]: another clustering algorithm that does not require $k$ beforehand.

We used a simple polynomial kernel of the form $\kappa(x, y) = (x \cdot y + 0.05)^3$ for the real-world data sets, and a dot product for the artificial set. Algorithm 3 was run with $\tau_1 = n \log(n/0.01)$ and $\tau_2 = k \log(k/0.01)$ mixing steps for first and second phases, respectively, and $C = R = 10$. The running time of our algorithm includes the time taken to heuristically search for $\lambda$ using the following BIC [14]:

$$BIC_k \triangleq \sum_{x \in P} \log \Pr(x|\{m_i\}_{i=1}^k, \sigma) - \frac{kd}{2} \log n,$$

where $\sigma$ is the average of each cluster's distortion, and $d$ is the dimension of the data set. The tuning procedure is the same as the one given at the end of the previous section, without using binary search.

## 4.1 Real-World Data Sets

The plots of the distortion and time for the LET set over the clustering iterations are given in Figure 1. Recall that KKM was run with the true number of clusters as its input, so one may expect it to perform relatively better, in terms of distortion and running time, than the other algorithms, which must compute $k$. The plots show that this is indeed the case, with our DPP-MC outperforming its competitors. Both DPP-Full and DPP-MAP require long running time for the eigen-decomposition of the similarity matrix. It is interesting to note that DPP-MAP does not perform better than a plain DPP-Full. We conjecture that this phenomenon is due to the approximate nature of the MAP inference.

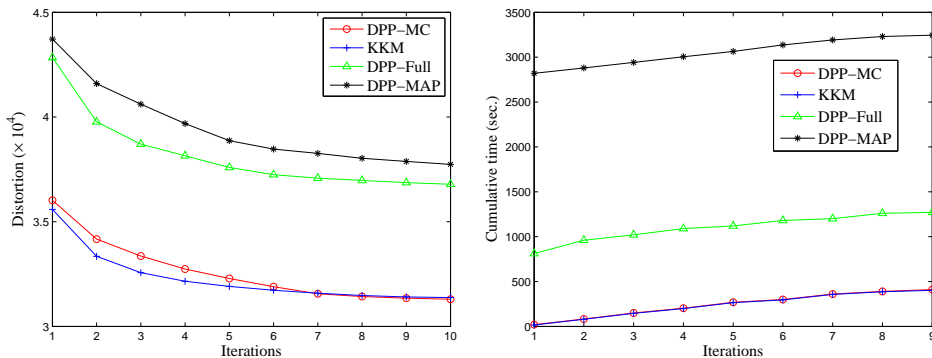

Figure 1: Distortion (left) and cumulated runtime (right) of the clustering induced by the competing algorithms on the LET set.

In Table 1, we give a summary of the DPP-based initialization procedures. The reported values are the immediate results of the initialization. For DPP-MC, the running time includes the automated $\lambda$ tuning. Taking this fact into account, DPP-MC was able to recover the true value of $k$ quickly.

In Figure 2, we show the same results on the PWC set. As in the previous case, DPP-MC exhibits the lowest distortion with the fastest running time. For this set, we have omitted the results for DPP-

| | DPP-MC | DPP-Full | DPP-MAP | DPP-MC | DPP-Full | DPP-MAP |
|---|---|---|---|---|---|---|
| Distortion | **36020** | 42841 | 43719 | **9.78** | 20.15 | 150 |
| Time (sec.) | **20** | 820 | 2850 | **15** | 50 | 220 |
| $k$ | **26** | 6 | 16 | **13** | 6 | 1 |
| $\lambda$ | 2 | - | - | 4 | - | - |

Table 1: Comparison among the DPP-based initializations for the LET set (left) and the PWC set (right).

MAP, as it yielded a degenereate result of a single cluster. Nevertheless, we give the final result in Table 1.

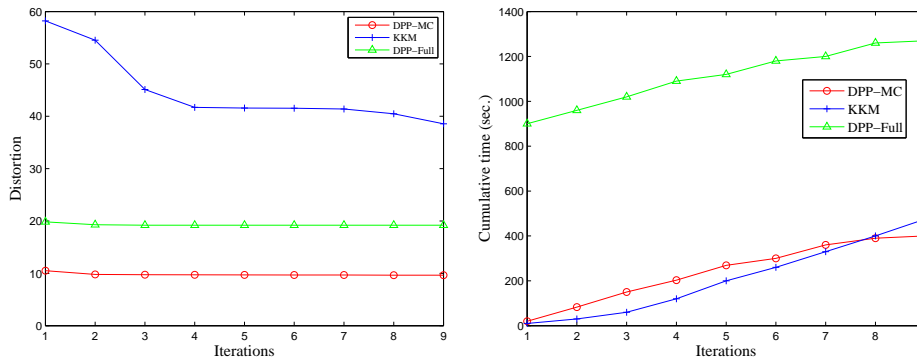

Figure 2: Distortion (left) and time (right) of the clustering induced by the competing algorithms on the PWC set.

## 4.2 Artificial Data Set

Finally, we present results on clustering the artificial MG set. In this set, we compare our algorithm with another clustering algorithm DBSCAN that does not require $k$ a priori. Due to page constraints, we summarize the result in Table 2.

| | DPP-MC | DBSCAN |
|---|---|---|
| Distortion | **6.127** | 35.967 |
| Time (sec.) | 416 | 60 |
| $k$ | 34 | 1 |

Table 2: Comparison among the DPP-based initializations for the PWC set.

Due to the merged configuration of the MG set, DBSCAN is not able to successfuly discover multiple clusters, and ends up with a singleton clustering. On the other hand, DPP-MC managed to find many distinct clusters in a way the distortion is lowered.

## 5 Discussion and Future Works

We have proposed a fast method for sampling from an $\epsilon$-close DPP distribution and its application to kernelized clustering. Although the exact computational complexity of sampling ($O(T^2 n \log(n/\epsilon))$) is not explicitly superior to the previous approach ($O(n^3)$), we emperically show that $T$ is generally small enough to account for our algorithm's efficiency. Furthermore, the extension to $k$-DPP sampling yields very fast speed-up compared to the previous sampling algorithm.

However, one must keep in mind that the mixing time analysis is for a single sample only: *i.e.*, we must mix the chain for each sample we need. For a small number of samples, this may compensate for the cubic complexity of the previous approach. For a larger number of samples, we must further

investigate the effect of sample correlation after mixing in order to prove long-term efficiency.

## Footnotes

*This work was submitted when the author was a graduate student at KAIST.

[1]Also known as $L$-ensembles.

[2]All proofs, including those of irreducibility of our chains, are given in the Appendix of the full version of our paper.

[3]The distance between $x$ and $y$ is defined as $\overline{\kappa(x,x) - 2\kappa(x,y) + \kappa(y,y)}$, for any positive semi-definite kernel $\kappa$

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
