[Supplementary Material]

# Supplementary Material

**Byungkon Kang**
SAIT, Yongin, Korea
bk05.kang@samsung.com

I would like to apologize for the following error in the published paper "Fast determinantal point process sampling with application to clustering".
The proofs for Theorem 1 and Theorem 2 rely on the wrong derivation of the Markov chains being couplings.
In the proof of Theorem 1, the following claim is made (verbatim):

> "The coupling process $(v_X, v_Y) \rightarrow (v_{X'}, v_{Y'})$ is simple: Assume that the position $v_X$ and $v_Y$ differs is $i$. If $u \neq i$, we set $v_{X'}$ according to Algorithm 1, and set $v_{Y'} = v_{X'}$. If $u = i$, $v_X$ and $v_Y$ are updated separately and independently using Algorithm 1."

However, the process of setting $v_{Y'} = v_{X'}$ violates the definition of a coupling. This is because $v_{Y'}$ should be updated from $v_Y$ by Algorithm 1 using the acceptance probability that depends on $Y$ and $Y'$ only. $v_{X'}$ is NOT derived from such probability, and thus this process does not qualify for a valid coupling. The same logic applies to the case of Theorem 2 as well.

Furthermore, [1] proves that the mixing time of an MCMC-based DPP sampling can be arbitrarily unbounded. Such negative results invalidate most of this work, hence readers should refrain from citing this work, or take caution when doing so.

It is possible that the proposed chain does yield "good enough" samples, which seems to be the case with the empirical result of this paper, but this may be due to the particular structure of the kernel matrix. In general, this does not hold. For the general form of the mixing time, please refer to theorems 1 and 2 of [2], where the individual acceptance probabilities are explicitly present in the bound on the mixing time.
The theorems provided in [2] may be applicable to Algorithm 1, but this still does not guarantee rapid mixing, since the mixing time bound can be arbitrarily large depending on the acceptance probabilities. It can, however, be used to derive an educated guess on the approximate mixing time. One might be able to use this guess to estimate the loss (*e.g.*, total variation between the current distribution and the target DPP distribution) of terminating the chain early. I leave this as a future work for interested readers.

**Remark**: I would like to thank Dr. Jennifer Gillenwater for her helpful discussions. The following portion of texts is the (erroneous) original text uploaded previously. It is left for your reference.

## 1 Path Coupling

**Definition 1** (Coupling [3]). *A coupling of a Markov chain $M$ is a joint process $(X_t, Y_t) \in \Omega \times \Omega$, such that*

$$\Pr(X_{t+1} = x' | X_t = x, Y_t = y) = P(x, x')$$
$$\Pr(Y_{t+1} = y' | X_t = x, Y_t = y) = P(y, y'),$$

*where $P(\cdot, \cdot)$ is the transition probability.*

The $\epsilon$-mixing time ($\tau(\epsilon)$) of $M$ is bounded via the following path coupling lemma.
**Path Coupling Lemma.** *Let $\delta$ be an integer valued metric defined on $\Omega \times \Omega$ which takes values in*

$\{0, 1, \cdots, D\}$. *Let the path set $R$ be a subset of $\Omega \times \Omega$ s.t. for all $(X_t, Y_t) \in \Omega \times \Omega$, there exists a path $X_t = Z_0, Z_1, \cdots, Z_r = Y_t$, where $(Z_l, Z_{l+1}) \in R$, and $\sum_{l=0}^{r-1} \delta(Z_l, Z_{l+1}) = \delta(X_t, Y_t)$. Suppose a Coupling $(X, Y) \to (X', Y')$ is defined on all pairs in $R$, s.t. for all $(X, Y) \in R$, $E[\delta(X', Y')] \le \beta\delta(X, Y)$ for some $\beta < 1$. Then, $\tau(\epsilon) \le \frac{\log(D\epsilon^{-1})}{1-\beta}$.*

## 2  Proof of Proposition 1

*Proof.* The proof of irreducibility proceeds as follows: Since the state space of this chain is the set of all characteristic vectors, each state can be represented as a 0-1 vector. For any two states, consider the indices $I = \{i\}$ at which the elements differ. Algorithm 1 assigns non-zero probabilities to individual element transitions $0 \to 1$ and $1 \to 0$ (*i.e.*, insertion and deletion probabilities). The probabilities are non-zero since no possible principle minor of $L$ can have a zero determinant, due to $L$ being positive definite.

Multiplying such probabilities for each $i \in I$ yields the probability of transitioning from one state to another. This means there is a non-zero probability of reaching an arbitrary state from any other states, hence making the chain irreducible.

Finally, to prove the stationary distribution, it suffices to prove the *detailed balance equation* since the chain is reversible by definition of the transition probability. The detailed balance equation is:

$$P_L(X)\Pr(X \to Y) = P_L(Y)\Pr(Y \to X), \forall X, Y \subseteq S,$$

for $X = Y \cup \{u\}$, without loss of generality. Then, the following holds:

$$\frac{\Pr(X \to Y)}{\Pr(Y \to X)} = \frac{p_u^-(X)/n}{p_u^+(Y)/n} = \frac{\det(L_Y)}{\det(L_X)}.$$

This is because exactly one of $p_u^+(Y)$ and $p_u^-(X)$ has to be 1, and the other less than 1. The last ratio is expanded as:

$$\frac{\det(L_Y)}{\det(L_X)} = \frac{Z\det(L_Y)}{Z\det(L_X)} = \frac{P_L(Y)}{P_L(X)},$$

where $Z$ is the normalization factor of the DPP $P_L$. Hence, the detailed balance equation and, consequently, our claim on the stationary distribution holds. □

## 3  Proof of Theorem 1

*Proof.* We prove the mixing time via the *path coupling* method [3].

The state distance metric we use is the Hamming distance: $\delta(v_X, v_Y) = \sum_{i=1}^{|S|} |v_X(i) - v_Y(i)|$. As per the transition setting of Algorithm 1, we designate the path set to be $R = \{(v_X, v_Y), \delta(v_X, v_Y) = 1\}$. The coupling process $(v_X, v_Y) \to (v_{X'}, v_{Y'})$ is simple: Assume that the position $v_X$ and $v_Y$ differs is $i$. If $u \ne i$, we set $v_{X'}$ according to Algorithm 1, and set $v_{Y'} = v_{X'}$. If $u = i$, $v_X$ and $v_Y$ are updated separately and independently using Algorithm 1.

Notice that $\delta(v_{X'}, v_{Y'}) \le 1$, and there are two possible cases when the distance is 1: 1) when $u \ne i$, or 2) when $u = i$ and both $v_X$ and $v_Y$ are updated. The probability of the former event is $1 - 1/n$. Assuming $X = Y \cup \{u\}$, the probability of the latter event is $p_u^+(Y)p_u^-(X)/n$, which expands to

$$\frac{p_u^+(Y)p_u^-(X)}{n} = \frac{1}{n}\min\left\{\frac{\det(L_Y)}{\det(L_X)}, \frac{\det(L_X)}{\det(L_Y)}\right\} \triangleq \frac{\alpha_{u,X}}{n}.$$

Hence, we have

$$E[\delta(v_{X'}, v_{Y'})] = 1 - \frac{1}{n}(1 - \alpha_{u,X}),$$

for a chosen $u$. Since this holds for any $u$, we upper-bound the final quantity by using $\alpha = \max_{u,X} \alpha_{u,X}$. We get the statement of the theorem by applying the path coupling lemma. □

# 4   Proof of Proposition 2

*Proof.* We first show the irreducibility of the chain. Here, the state space is the set of all 0-1 vectors of length $n$ with exactly $k$ 1's. For any two states $v$ and $u$, consider the index set $I = \{i\}$ at which the elements of the two states differ. We claim that $|I|$ is always even. To see why this holds, suppose there are an odd number of mismatches. Then, one of the states must hold extra 1's at some positions $J = \{j | j \notin I\}$ to account for the unequal number of 1's in positions $I$. However, the other state must also have 1's at its positions $J$ since the only mismatches are in $I$. This will make one state have more than $k$ 1's, leading to a contradiction.

Since there are even mismatches, we can pair up the indices in $I$. We do so, so that for each pair $(i_1, i_2), v(i_1) \neq v(i_2)$ (the same holds for $u$ by definition of $I$). Then, transitioning from $v$ to $u$ (and vice versa) is done by swapping the elements at the indices of each pair; *i.e.*, $v(i_1) \leftrightarrow v(i_2), \forall (i_1, i_2)$. Each such swap, and hence the overall transition, has non-zero probability by definition of our chain and the positive-definiteness of $L$. Thus, our chain is irreducible.

Proving the detailed balance equation is similar to Proposition 1. According to Algorithm 2, we have:

$$\Pr(X \to X')/\Pr(X' \to X) = \det(L_X)/\det(L_{X'}),$$

because exactly one of $p = \det(L_X)$ and $p^{-1}$ is 1, and the other less than 1. Expanding the final ratio as $Z \det(L_X)/Z \det(L_{X'})$ gives $P_L(X)/P_L(X')$, completing the proof. $\qquad\square$

# 5   Proof of Theorem 2

*Proof.* We again use the path coupling technique. The state space is $\Omega = \{v_Z || Z| = k\}$. Let the path set be $R = \{(v_A, v_B) | \delta(v_A, v_B) \leq 2\}$, under the Hamming metric $\delta$. Define the coupling $(v_X, v_Y) \to (v_{X'}, v_{Y'})$ as follows:

Let $s$ and $t$ be the indices at which $v_X$ and $v_Y$ differ. By definition of the path set $R$, these will be the only positions of difference. *i.e.*, $v_X(s) = 1, v_Y(s) = 0$ and $v_X(t) = 0, v_Y(t) = 1$, without loss of generality. Construct a bijection $g : [n] \mapsto [n]$ such that

$$g(x) = \begin{cases} s & \text{if } x = t \\ t & \text{if } x = s \\ x & \text{otherwise.} \end{cases}$$

Choose $i \in X$ and $j \in R \backslash X$ u.a.r. Compute $p_1$ and $p_2$ from Equation 5 for $v_X$ and $v_Y$, respectively. Then, update $v_{X'}$ with $i, j$, and $v_{Y'}$ with $g(i), g(j)$ independently using Algorithm 2, if at least one of $i = t$ or $j = s$ holds. Otherwise, update $v_{X'}$ as Algorithm 2 and set $v_{Y'} \leftarrow v_{X'}$.

To compute $E[\delta']$, where $\delta' \triangleq \delta(v_{X'}, v_{Y'})$, there are four cases to consider (note that $p_1$ and $p_2$ differ from case to case).

- $i = s$ and $j = t$ (w.p. $\frac{1}{k(n-k)}$): With probability $\alpha_1 = p_1 p_2$, both updates for $v_{X'}$ and $v_{Y'}$ are accepted, resulting in $\delta' = 2$. Otherwise, $\delta' = 0$.

- $i = s$ and $j \neq t$ (w.p. $\frac{n-k-1}{k(n-k)}$): With probability $\alpha_2 = 1 - p_1 p_2$, $\delta' = 2$.

- $j \neq s$ and $j = t$ (w.p. $\frac{k-1}{k(n-k)}$): With probability $\alpha_3 = 1 - p_1 p_2$, $\delta' = 2$.

- Otherwise (w.p. $\left(1 - \frac{1}{k}\right)\left(1 - \frac{1}{n-k}\right)$): $\delta' = 2$ with probability 1 (due to coupling).

Letting $\alpha \triangleq \min\{\alpha_1, \alpha_2, \alpha_3\}$ over all possible $p_1, p_2$, the expected value is:

$$
\begin{aligned}
E[\delta'] =& \delta(X,Y)\left(\frac{\alpha_1 + \alpha_2(n-k-1) + \alpha_3(k-1)}{k(n-k)} + \left(1 - \frac{1}{k}\right)\left(1 - \frac{1}{n-k}\right)\right) \\
\leq& \delta(X,Y)\left(1 - \frac{(1-\alpha)(n-1)}{k(n-k)}\right) \\
\leq& \delta(X,Y)\left(1 - \frac{1-\alpha}{k}\right) \quad (k > 1).
\end{aligned}
$$

We get the statement by applying the path coupling lemma. $\qquad\square$