[Reviews · NeurIPS 2013]

Submitted by Assigned_Reviewer_1

The goal of this paper is to build a markov chain that will sample from a determinantal point process. One that mixes rapidly, and improves on the O(n^3) direct computation. One benefit is that as the set of elements in the DPP changes, there is no expensive eigenvalue decomposition. The fast algorithm is achieved with the realization that the algorithm doesn't require the computation of matrix determinants, but the ratio of determinants. The authors then apply the DPP as a method for choosing the number of clusters in k-means.

This paper is exceptionally well written and easy to read. The idea for sampling from a DPP using a markov chain, and simplifying the computational complexity of the necessary matrix computations strikes me as rather clever. I am not familiar enough with different applications of DPPs to be able to comment on the significance.

I'm a little disappointed to see this sophisticated & interesting idea applied to k-means. The real beauty & utility of k-means is its simplicity, and that's also it's drawback. k-means is exactly equivalent to fitting a gaussian mixture model using the E-M algorithm, under the assumption that the variance of each gaussian component is diagonal. And k-means can fail miserably when this variance assumption is wrong, even when the number of clusters is known. I don't see how applying DPP to k-means can fix this bigger issue.

I'm also not fully convinced that the cardinality penalty will actually handle outliers. The penalty controls how big Y gets, but if outliers add more diversity then wouldn't the outliers just be replacing other points in the smaller set Y?
Summary: This is a well written paper that uses a clever idea to speed up sampling from a DPP. I'm not convinced that using a DPP to choose the number of clusters in k-means is very helpful.

Submitted by Assigned_Reviewer_4

This paper provides an efficient sampling algorithm for DPPs. Though their algorithm is not explicitely better than the standard sampling algorithm for DPPs, in practice their algorithm outperforms the standard algorithm. They also consider real world and synthetic applications involving clustering, there they use the DPPs to give a hueristic for sampling.

I thnk the application of Markov chain mixing techniques to sampling DPPs is nice, and the faster sampling algorithm could be useful in many applications. Overall the paper also seems well written.

However, I am not fully convinced on the application of DPPs for clustering. Though DPPs seem to model diversity in many real world applications, it is not intuitive that the DPPs are in general a better model for clustering. Moreover, even using DPPs there are no guarantees for clustering and the procedure seems hueristic. In that context, it is not clear why the DPPs will necessarily offer a better solution for clustering. I think this should be clarified better in the paper. Also, are there some other applications where one might want to sample from a DPP?

A recent relevant paper at UAI this year, is "Determinantal Clustering Processes - A Nonparametric Bayesian Approach to Kernel Based Semi-Supervised Clustering". Though not exactly the same problem, the paper is closely related to this one.

* Edits after the author rebuttal period and discussions
I feel that the paper is well written, though only incremental in my opinion. The main cons are that the application of the sampling algorithm to k-means seems very heuristic, and hence I will stick to my score of 6 (i.e borderline accept). I think the paper can be significantly improved, if the authors could point out some other applications of DPP sampling, where their algorithms will imply significant improvement in running time.
Summary: This paper provides a novel sampling algorithm for DPPs based on markov chain mixing. Emperically and theoretically the authors motivate their sampling algorithm and consider an application of clustering. I do not see enough intuition for using DPPs for clustering and also would like to see more general application of these sampling techniques.
Author Feedback

Author rebuttal: We would like to thank the reviewers for their time and constructive criticisms.
We agree with the concern raised that the application to clustering is somewhat simple. Our initial goal was to apply our algorithm to a multi-agent search problem, where a small number of agents must lead other non-leader agents to achieve maximum coverage of the search area. We concluded that this problem was related to clustering, and in the interest of time, we have decided to first demonstrate success on a more abstract clustering problem. In the subsequent works, we will focus mainly on our original aim.

Some short responses to other questions:
*The cardinality penalty is mainly intended to keep the clustering model simple (ie, small # of clusters)
*The heuristic intuition of DPP-clustering is that diverse points should better serve as representatives of each cluster